# *BRCA1/2* Mutation Testing in Patients with HER2-Negative Advanced Breast Cancer: Real-World Data from the United States, Europe, and Israel

**DOI:** 10.3390/cancers14215399

**Published:** 2022-11-02

**Authors:** Reshma Mahtani, Alexander Niyazov, Bhakti Arondekar, Katie Lewis, Alex Rider, Lucy Massey, Michael Patrick Lux

**Affiliations:** 1Miami Cancer Institute, 8900 Kendall Drive, Miami, FL 33176, USA; 2Patient and Health Impact, Pfizer Inc., 235 42nd St., New York, NY 10017, USA; 3Patient Health and Impact, Pfizer Inc., 500 Arcola Road, Collegeville, PA 19426, USA; 4Oncology Franchise, Adelphi Real World, Adelphi Mill, Cheshire, Bollington SK10 5JB, UK; 5Department of Gynecology and Obstetrics, Kooperatives Brustzentrum Paderborn, Frauenklinik St. Louise, Frauenklinik St. Josefs, Salzkotten Husener Straße 81, 33098 Paderborn, Germany

**Keywords:** advanced breast cancer, breast cancer susceptibility genes 1 and 2, genetic testing, human epidermal growth factor receptor 2—negative, poly(adenosine diphosphate-ribose) polymerase inhibitors, real-world

## Abstract

**Simple Summary:**

Poly(adenosine diphosphate-ribose) polymerase inhibitors have recently been shown to be effective for patients with human epidermal growth factor receptor 2—negative (HER2−) advanced breast cancer (ABC) who have a germline mutation in their breast cancer susceptibility gene 1 or 2 (*BRCA1/2*mut). This study evaluated differences in patient demographics, clinical characteristics, and *BRCA1/2*mut testing within the United States (US), European Union 4 (EU4; France, Germany, Italy, and Spain), and Israel in a real-world patient population with HER2− ABC. In the US, EU4, and Israel, 73%, 42%, and 99% of patients were tested for *BRCA1/2*mut, respectively. In the US and the EU4, patients who were not tested versus tested for *BRCA1/2*mut were more likely to have hormone receptor–positive (HR+)/HER2− ABC than triple-negative breast cancer, less likely to have a known family history of *BRCA1/2-*related cancer and were older. Efforts should be made to improve *BRCA1/2* testing rates in the US and Europe.

**Abstract:**

Poly(adenosine diphosphate-ribose) polymerase inhibitors are approved to treat patients harboring a germline breast cancer susceptibility gene 1 or 2 mutation (*BRCA1/2*mut) with human epidermal growth factor receptor 2—negative (HER2−) advanced breast cancer (ABC). This study evaluated differences in patient demographics, clinical characteristics, and *BRCA1/2*mut testing within the United States (US), European Union 4 (EU4; France, Germany, Italy, and Spain), and Israel in a real-world population of patients with HER2− ABC. Oncologists provided chart data from eligible patients from October 2019 through March 2020. In the US, EU4, and Israel, 73%, 42%, and 99% of patients were tested for *BRCA1/2*mut, respectively. In the US and the EU4, patients who were not tested versus tested for *BRCA1/2*mut were more likely to have hormone receptor—positive (HR+)/HER2− ABC (US, 94% vs. 74%, *p* < 0.001; EU4, 96% vs. 78%, *p* < 0.001), less likely to have a known family history of *BRCA1/2-*related cancer (US, 6% vs. 19%, *p* = 0.002; EU4, 10% vs. 28%, *p* < 0.001), and were older (US, 68.9 vs. 62.5 years, *p* < 0.001; EU4, 66.7 vs. 58.0 years, *p* < 0.001). Among tested patients, genetic counseling was received by 45%, 53%, and 98% with triple-negative breast cancer, and 36%, 36%, and 98% with HR+/HER2− ABC in the US, EU4, and Israel, respectively. Efforts should be made to improve *BRCA1/2* testing rates in the US and Europe.

## 1. Introduction

An estimated 5% to 10% of breast cancers are caused by a genetic predisposition resulting from a mutation in a gene that increases the risk of breast cancer [1]. The genes most commonly affected in hereditary breast cancer and ovarian cancer are breast cancer susceptibility genes 1 and 2 (*BRCA1/2*) [2]. Approximately 3% to 6% of all breast cancer cases are caused by a *BRCA1/2* mutation (*BRCA1/2*mut) [3,4,5], and women with a genetic *BRCA1/2*mut have a cumulative 45% to 66% risk of developing breast cancer by 70 years of age [2]. Accordingly, genetic testing for breast cancer susceptibility has become an important part of disease management [1].

Tumors with a *BRCA1/2*mut are highly sensitive to inhibition of poly(ADP-ribose)polymerase (PARP) [6]. In 2018, the PARP inhibitors (PARPi) olaparib and talazoparib were approved by the US Food and Drug Administration (FDA) for treatment of patients with human epidermal growth factor receptor 2—negative (HER2−) advanced breast cancer (ABC) harboring a germline *BRCA1/2*mut (g*BRCA1/2*mut) and are now available in many countries for the treatment of g*BRCA1/2*mut HER2− ABC [7,8,9]. The approvals were based primarily on findings from the OlympiAD and EMBRACA randomized, open-label trials, which demonstrated a significantly improved progression-free survival, manageable adverse event profile, and improved patient-reported outcomes in patients with g*BRCA1/2*mut HER2− ABC who received olaparib or talazoparib compared with patients who received physician’s choice of chemotherapy (OlympiAD: olaparib versus capecitabine, vinorelbine, or eribulin; EMBRACA: talazoparib versus capecitabine, vinorelbine, eribulin, or gemcitabine) [10,11,12,13,14]. These findings underscore that, in addition to hormone receptor (HR) status, HER2 status, and programmed death ligand 1 (PD-L1) status in triple-negative breast cancer (TNBC), information about *BRCA1/2*mut status is also an essential factor in determining choice of therapy.

With the approval of PARPi for germline (though not somatic) mutations, and the potential for effective therapeutic intervention in patients with a *BRCA1*/*2*mut, national and international guidelines have broadened eligibility criteria for g*BRCA1/2*mut testing [15,16]. The present analyses evaluated differences in patient demographics and clinical characteristics in a real-world population of patients with HER2− ABC to identify potential factors contributing to physicians’ decisions to test for a *BRCA1*/*2*mut within the United States, European Union 4 (EU4; France, Germany, Italy, and Spain), and Israel. We also evaluated whether, and when, patients had undergone genetic counseling for *BRCA1/2*mut testing.

## 2. Methods

### 2.1. Data Source and Study Design

Data were obtained from the Adelphi Real World Disease Specific Programme (DSP^TM^) for ABC, and the study was conducted from October 2019 through March 2020 in the United States, the EU4, and Israel. DSPs are large, multinational, point-in-time surveys of physicians and their patients presenting in a real-world clinical setting that assess disease management, disease-burden impact, and associated treatment effects [17].

Participating physicians were medical oncologists evaluating ≥5 patients with ABC per month, were actively involved in treating patients, and were recruited by local study teams. Physicians provided patient record forms (PRFs) for the next 8 eligible consulting patients: 4 patients receiving first-line advanced treatment and 4 receiving second- or later-line advanced treatment. Eligible patients were ≥18 years of age with stage IIIb to IV HER2− breast cancer and receiving therapy for ABC at the time of data collection; patients participating in a clinical trial were not eligible. Physicians reported on biomarker testing, including but not limited to homologous recombination repair genes, HER2, PD-L1, progesterone and estrogen receptor, PIK3CA, and *BRCA1/2*, and were asked the proportion of patients tested and the proportion of positive tests. Physicians were asked to report if testing was performed on blood, saliva, or buccal samples, and this information was used to confirm that *BRCA1/2*mut testing was germline. For US-based patients, this was also verified by inquiring the name of the laboratory where the testing was performed, whereas data for laboratory confirmation of test type were not available for the EU4 or Israel (Figure 1).

The PRF included detailed questions on patient demographics, clinical assessments and outcomes, adverse events experienced at the time of data collection, treatment history, and physician-rated satisfaction with treatment. Physicians completed the PRFs using patient medical records as well as clinical judgment and diagnostic skills consistent with their decision-making process during routine clinical practice. Each patient with a PRF was invited to complete an optional patient form by pen and paper independently of their physician immediately after the consultation. The patient form included questions on their education, employment status, input to treatment decisions, and current disease status, as well as patient-reported outcome questionnaires that assessed their quality of life.

Patients provided informed consent for use of their anonymized and aggregated data for research and in scientific publications. Data were aggregated and de-identified before receipt by Adelphi Real World. The study was conducted in accordance with the Declaration of Helsinki and was approved by the Western Institutional Review Board (study protocol AG8643). Data collection was undertaken in line with European Pharmaceutical Market Research Association guidelines [18] and as such did not require ethics committee approval. Each survey was administered in full accordance with relevant legislation at the time of data collection, including the US Health Insurance Portability and Accountability Act of 1996 [19].

### 2.2. Outcomes and Measures

*BRCA1/2* mutation testing rates and characteristics of patients undergoing testing were stratified by the type of test performed: any *BRCA1/2*mut, g*BRCA1/2*mut with or without a somatic *BRCA1/2*mut (g +/− s*BRCA1/*2mut), s*BRCA1/*2mut-only, unknown *BRCA1/2*mut (i.e., the physician was not aware of testing results, or it could not be verified if mutations were somatic or germline), and no *BRCA1/2*mut testing. Results were also stratified by HR status (i.e., HR+/HER2− or TNBC), practice setting, age, and family history of *BRCA1/2*-related cancer, and between-group comparisons were performed to identify possible factors that may have contributed to the decision to test patients within each region. Rates and timing of genetic counseling (i.e., before and/or after *BRCA1/2*mut testing) within each *BRCA1/2*mut testing group were also determined. Genetic counseling was performed by a geneticist or the treating physician.

### 2.3. Statistical Analysis

Descriptive summary statistics, including the mean, standard deviation, median, and range, were calculated for continuous variables. Frequency counts and percentages were calculated for categorical variables. Differences in demographics and clinical characteristics among *BRCA1/2*mut testing status groups were analyzed by Student’s *t*-tests or Fisher exact tests. Values with *p* < 0.05 were considered statistically significant. A binomial exact test was performed to compare patients who received versus did not receive genetic counseling. Percentages and 95% CIs were reported; 95% CIs that did not cross 50%, or 0.50, indicated a significant difference (*p* < 0.05). Missing data were not imputed; thus, the sample size varied among variables assessed and is reported separately for each analysis. Analyses were performed with IBM^®^ SPSS^®^ Data Collection Survey Reporter Version 6 or later (International Business Machines Corp., Armonk, NY, USA) and STATA version 16.1 or later (StataCorp, College Station, TX, USA).

## 3. Results

### 3.1. BRCA1/2 Mutation Testing in the United States

Physicians completed PRFs for 407 US patients. Patients had a mean age of 64.2 years, 6% (*n* = 26) were premenopausal, 15% (*n* = 63) had a known family history of *BRCA1/2*-related cancer, 80% (*n* = 325) had HR + /HER2− disease, and 20% (*n* = 82) had TNBC. US patient characteristics stratified by *BRCA1/2*mut testing status are shown in Table 1. Overall, 73% (*n* = 298) of patients were tested for any type of *BRCA1/2*mut (germline, somatic, or unknown); among these, 47% (*n* = 190) received a g +/− s*BRCA1/*2mut test, 18% (*n* = 75) received an s*BRCA1/2*mut-only test, and 8% (*n* = 33) received an unknown type of *BRCA1/2*mut test. Those who were not tested for any *BRCA1/2*mut were significantly older than those who were tested (68.9 vs. 62.5 years; *p* < 0.001) and significantly less likely to be employed (18% vs. 33%; *p* = 0.003), premenopausal (2% vs. 8%; *p* = 0.022), have a family history of *BRCA1/2*-related cancer (6% vs. 19%; *p* = 0.002), have TNBC (6% vs. 26%; *p* < 0.001), or be tested in an academic setting (28% vs. 41%; *p =* 0.021) versus those who were tested.

Evaluating associations between *BRCA1/2*mut testing rates and HR+/HER2− and TNBC subtypes among US patients indicated that those with TNBC were tested for a g +/− s*BRCA1/*2mut at significantly higher rates compared with patients with HR+/HER2− disease (61% vs. 43%; *p =* 0.004; Table 2). s*BRCA1/2*mut-only testing rates were similar between patients with TNBC and those with HR+/HER2− disease (20% vs. 18%; *p* = 0.75).

Among patients with HR+/HER2− disease, fewer patients received any *BRCA1/2*mut testing in a community medical center compared with those in an academic medical center (64% vs. 75%, *p* = 0.048; Table 3). Those receiving treatment in an academic medical center were significantly more likely to receive g +/− s*BRCA1/*2mut testing but less likely to receive s*BRCA1/*2mut-only testing compared with those receiving care in a community medical center (g +/− s*BRCA1/*2mut, 54% vs. 37%, *p* = 0.004; s*BRCA1/*2mut-only, 12% vs. 22%, *p* = 0.039). Testing rates for each of the *BRCA1/2*mut testing groups among patients with TNBC were not significantly different across academic and community medical centers.

Among patients with HR+/HER2− ABC, overall *BRCA1/2*mut testing rates were lower for those who had no known family history of *BRCA1/2*-related cancer compared with those who did have a family history (67% vs. 84%, *p* = 0.030; Table 4). Among patients with TNBC, testing rates across all testing groups were not significantly different in patients with and without a known family history of *BRCA1/2*-related cancer.

When stratified by age group, *BRCA1/*2mut testing rates among patients with HR+/HER2− ABC declined with age, with 100%, 92%, 75%, and 60% of patients < 45, 45 to 54, 55 to 64, and ≥65 years of age, respectively, having any type of *BRCA1/2*mut test (Figure 2A). Among patients with TNBC, testing rates only slightly declined with age, with all patients < 55 years of age, 95% of patients 55 to 64 years of age, and 85% of patients ≥ 65 years of age having received a *BRCA1/*2mut test.

Among US patients with HR+/HER2− ABC tested for any *BRCA1/*2mut, 36% received genetic counseling (73 [91%] from a genetic counselor and 8 [10%] from the treating physician), 52% did not receive counseling (received vs. did not receive counseling: binomial test proportion 0.41 [95% CI, 0.34−0.48]), and, for 12% of patients, it was unknown if they received genetic counseling (Figure 3A). Approximately equal percentages of patients within this group received counseling before (13%), after (13%), or both before and after (9%) genetic testing; for 1% of patients, the timing of counseling was unknown (Figure 3A). The g +/− s*BRCA1/*2mut and s*BRCA1/*2mut-only testing subgroups had similar percentages of patients who received genetic counseling, 34% and 47%, respectively, but varied by the distribution of time points at which counseling was received. Among the patients with TNBC tested for any *BRCA1/2*mut, 45% received genetic counseling (88% from a genetic counselor and 12% from the treating physician), 37% did not receive counseling (received vs. did not receive counseling: binomial test proportion 0.54 [95% CI, 0.42−0.68]), and, for 18% of patients, it was unknown if they received genetic counseling (Figure 3A). As with the patients with HR+/HER2− ABC tested for any *BRCA1/2*mut, similar percentages of the patients with TNBC tested for any *BRCA1/2*mut received counseling before (16%), after (16%), or both before and after (13%) genetic testing (Figure 3A).

### 3.2. BRCA1/2 Mutation Testing in the European Union 4

Physicians completed PRFs for 1926 EU4 patients. Patients had a mean age of 63.1 years, 8% (*n* = 151) were premenopausal, 17% (*n* = 337) had a known family history of *BRCA1/2*-related cancer, 88% (*n* = 1703) had HR+/HER2− disease, and 12% (*n* = 223) had TNBC. EU4 patient characteristics stratified by *BRCA1/2*mut testing status are shown in Table 5. Overall, 42% (*n* = 805) of the patients were tested for any type of *BRCA1/2*mut; among these, 27% (*n* = 528) received a g +/− s*BRCA1/*2mut test, 10% (*n* = 186) received an s*BRCA1/2*mut-only test, and 5% (*n* = 91) received an unknown type of *BRCA1/2*mut test. Those who were not tested for a *BRCA1/2*mut were significantly older than those who were tested (66.7 vs. 58.0 years; *p* < 0.001) and significantly less likely to be employed (11% vs. 26%; *p* < 0.001), be premenopausal (3% vs. 15%; *p* < 0.001), have a family history of *BRCA1/2*-related cancer (10% vs. 28%; *p* < 0.001), have TNBC (4% vs. 22%; *p* < 0.001), or be tested in an academic setting (52% vs. 61%; *p* < 0.001).

Patients in the EU4 with TNBC were tested for a g +/− s*BRCA1/2*mut at significantly higher rates compared with patients with HR+/HER2− disease (57% vs. 24%; *p* < 0.001); the same was true for s*BRCA1/2*mut-only testing (14% vs. 9%; *p* = 0.029; Table 2). For patients with HR+/HER2− disease and those with TNBC, patients in academic medical centers were more likely to receive any *BRCA1/2*mut testing compared with those treated in community medical centers (HR+/HER2−, 41% vs. 33%; TNBC, 89% vs. 65%; both *p* < 0.001; Table 3). Considering family history, patients in the EU4 with HR+/HER2− ABC who had no known *BRCA1/2*-related family history were tested for any *BRCA1/2*mut at significantly lower rates than those who did have a family history (32% vs. 62%; *p* < 0.001; Table 4). For patients with TNBC, testing rates were only significantly lower for any *BRCA1/2*mut testing among those with no family history (75% vs. 89%; *p* = 0.023).

*BRCA1/2* mutation testing rates among patients in the EU4 with HR+/HER2− ABC declined with age, with 89%, 53%, 41%, and 25% of patients < 45, 45 to 54, 55 to 64, and ≥65 years of age, respectively, receiving any type of *BRCA1/2*mut testing (Figure 2B). The same trend was observed among patients with TNBC, although, as noted, testing rates were generally lower among patients with HR+/HER2− disease compared with patients with TNBC.

Among EU4 patients with HR+/HER2− disease tested for any *BRCA1/2*mut, 36% received genetic counseling (177 [77%] from a genetic counselor and 57 [25%] from the treating physician), 60% did not receive counseling (received vs. did not receive counseling: binomial test proportion 0.38 [95% CI, 0.34–0.42]) and, for 4% of patients, it was unknown if they received counseling. Among the patients with TNBC tested for any *BRCA1/2*mut, 53% received genetic counseling (73 [79%] from a genetic counselor and 23 [25%] from the treating physician), 41% did not receive counseling (received vs. did not receive counseling: binomial test proportion 0.56 [95% CI, 0.48–0.64]) and, for 6% of patients, it was unknown if they received counseling. In patients with HR+/HER2− disease and those with TNBC, counseling was most often received before genetic testing (Figure 3B). Within each population, the percentage of patients who received genetic counseling was lowest among those tested for an s*BRCA1/2*mut only.

### 3.3. BRCA1/2 Mutation Testing in Israel

Physicians completed PRFs for 194 Israeli patients. Patients had a mean age of 56.7 years, 27% (*n* = 52) were premenopausal, 68% (*n* = 131) had a known family history of *BRCA1/2*-related cancer, 73% (*n* = 141) had HR+/HER2− disease, and 27% (*n* = 53) had TNBC. Overall, 99% (*n* = 192) of the patients were tested for any type of *BRCA1/2*mut; among these, 96% (*n* = 186) received a g +/− s*BRCA1/*2mut test, 2% (*n* = 3) received an s*BRCA1/2*mut-only test, and 2% (*n* = 3) received an unknown type of the *BRCA1/2*mut test. No significant differences in patient characteristics were observed among those who were tested for a *BRCA1/2*mut compared with those who were not. All patients received treatment at an academic medical center.

As expected, based on the nearly ubiquitous nature of *BRCA1/2*mut testing among Israeli patients, no significant differences were seen in testing rates by HR subtypes (Table 2) or *BRCA1/2*-related family history (Table 4). When stratified by HR subtype and age group, all patients received *BRCA1/2*mut testing, except for 2 with HR+/HER2− disease who were ≥65 years of age (Figure 2C). Nearly all Israeli patients who received *BRCA1/2*mut testing (98% for both HR+/HER2− and TNBC) also received genetic counseling, with most patients (77% of patients with HR+/HER2− and 92% of those with TNBC) receiving counseling after genetic testing (Figure 3C). All patients who received genetic counseling received it from a genetic counselor.

## 4. Discussion

Based on the efficacy of PARPi demonstrated in clinical trials and their subsequent approval for treatment of patients with g*BRCA1/2*mut HER2− ABC, guidelines on testing for g*BRCA1/2*mut have expanded to include new therapeutic indications in addition to clinical criteria such as patients diagnosed at an early age and patients with a strong family history (e.g., a first-degree relative diagnosed with breast cancer at an early age or with TNBC, two or more close relatives with breast cancer at any age, two or more close blood relatives with breast, pancreatic or prostate cancer at any age or a known *BRCA1/2* mutation in the family) [10,11,20]. The National Comprehensive Cancer Network Clinical Practice Guidelines in Oncology (NCCN Guidelines^®^) now recommend testing for a g*BRCA1/2*mut in all patients with recurrent or metastatic breast cancer to identify candidates for PARPi treatment [20]. The European Society for Medical Oncology (ESMO) international consensus guidelines recommend that patients with ABC be tested for a g*BRCA1/2*mut “as early as possible” [9].

This study used the Adelphi Real World DSP to evaluate *BRCA1/2*mut testing rates and related characteristics among patients with HER2− ABC in the United States, the EU4, and Israel during October 2019 to March 2020. We had previously assessed *BRCA1/2*mut testing rates in the United States and the EU5 (France, Germany, Italy, Spain, and the United Kingdom) in 2015 and 2017 to provide a historical baseline for *BRCA1/2*mut testing [21]. Average rates of testing for any *BRCA1/2*mut in 2015–2017 for patients with HR+/HER2− ABC and those with TNBC were 43% and 72%, respectively, in the United States and 18% and 33%, respectively, in the EU5. Testing rates were substantially higher in the current study; testing rates for any *BRCA1/2*mut in patients with HR+/HER2− ABC and those with TNBC were 68% and 93%, respectively, in the United States and 37% and 78%, respectively, in the EU4. The FDA approval in 2018 and the subsequent European Medicines Agency authorization of PARPi for the treatment of patients with HER2− ABC likely contributed to the increase in *BRCA1/2*mut testing rates from 2015 and 2017 to 2019 and 2020. Despite the increased rates of *BRCA1/2*mut testing in the current study, g*BRCA1/2*mut testing rates were still relatively low among some patient groups. Testing rates for patients with HR+/HER2− ABC were lower in both the United States and the EU4 compared with patients with TNBC, with only 37% of patients with HR+/HER2− ABC in the EU4 being tested for any *BRCA1/2*mut. The relatively higher rates of testing in patients with TNBC likely reflects the increased awareness of the prevalence of g*BRCA1/2*mut among these patients [22]. However, a substantial percentage of patients with TNBC, particularly in the EU4 (49 of 223 [22%]), were not tested for any *BRCA1/2*mut, underscoring the need for testing to inform treatment decisions for patients with such limited options [22].

*BRCA1/2* mutation testing rates in older patients were also relatively low in both the United States and the EU4, particularly for those with HR+/HER2− ABC. For example, among patients with HR+/HER2− ABC tested for any *BRCA1/2*mut in the United States, 60% of those who were ≥65 years of age and 100% of those <45 years of age were tested; in the EU4, only 25% of patients ≥ 65 years of age were tested, while 89% of those <45 years of age were tested. Testing rates in the United States and the EU4 were also generally lower among those who were postmenopausal, had no known *BRCA1/2*-related family history of cancer, and were treated in community medical centers (vs. academic medical centers). These findings highlight the need for increased g*BRCA1/2*mut testing, with efforts specifically concentrated among patients with these demographic or clinical characteristics, to aid in identification of patients eligible for PARPi treatment.

There were also appreciable numbers of patients in the United States (18%) and the EU4 (10%), but not Israel, who received s*BRCA1/2*mut-only testing. Although patients with metastatic breast cancer with an s*BRCA1/2*mut have been shown to respond to PARPi [23,24], PARPi have not been approved to treat this patient group. The ESMO international consensus guidelines indicate that the therapeutic implications of s*BRCA1/2*mut in patients with breast cancer need further evaluation and should not be used for decision-making in clinical practice [9]. Because patients who were enrolled in clinical trials were excluded from this study, the reason some patients received s*BRCA1/2*mut-only testing is uncertain, but it is possible that they provided tissue samples for experimental studies.

*BRCA1/2* mutation testing rates were notably higher in Israel compared with the United States and the EU4. This is likely related to the high percentages of Israeli patients with Ashkenazi Jewish ethnicity, which is associated with an approximately 10-fold increase in the prevalence of *BRCA1/2*mut relative to the general population (approximately 2.0% vs. 0.2%) [25]. Possibly related to this risk factor, the percentage of patients with *BRCA1/2*-related family history was much higher in Israel (68%) compared with the United States or EU4 (15% and 17%, respectively), and the mean age among Israeli patients was lower compared with patients in the United States or EU4 (56.7 vs. 64.2 and 63.1 years, respectively). In addition, the proportion of patients who were premenopausal was much higher in Israel (27%) compared with the United States (6%) and EU4 (8%). An awareness that breast cancer incidence is higher in younger patients (i.e., premenopausal patients) who carry a *BRCA1/2*mut may have also contributed to the high rate of *BRCA1/2*mut testing in Israel. Although we did not identify patients of Ashkenazi Jewish ethnicity in the United States or EU4, rates of *BRCA1/2*mut testing among Ashkenazi Jews in the United States and the EU4 may be higher than the general population.

The study also identified considerable gaps in genetic counseling among patients tested for a *BRCA1/2*mut in the United States and the EU4, particularly in patients with HR+/HER2− ABC, where less than half of those who were tested received genetic counseling. Genetic counseling is important because it informs patients and their family members not only about genetic predisposition but also different therapeutic strategies for treatment of their cancer. The ESMO international consensus guidelines recommend that genetic counseling be provided to patients with ABC and their families if a pathogenic germline mutation is identified [9]. A rapidly increasing demand for genetic counseling resulting from the recent expansion of indications of PARPi among patients with ABC, as well as patients with ovarian and prostate cancer, with a deleterious *BRCA1/2*mut may be contributing to the relatively low rate of patients receiving genetic counseling observed in this study. Our findings indicate that improved awareness of and access to genetic counseling is needed in the United States and the EU4. In Israel, as with g*BRCA1/2*mut testing, essentially all patients received genetic counseling; most (up to 92%) received posttest counseling only. In Israel, as well as in Germany, genetic counseling is a legal requirement as a part of genetic testing [26]. Furthermore, Israel has one of the highest levels of genetic counselors per capita, second only to Cuba and the United States [27]. These factors, along with the high level of awareness and genetic testing, likely contribute to the high rate of genetic counseling in Israel.

The strengths of this study include the use of real-world data, which are important for informing patient care [28], across a large patient population spanning multiple countries. The generalizability of study results may be limited in that the DSP only includes data from physicians willing to take part; furthermore, patients may not be fully representative of the broader patient population because data were more likely to be collected from patients who frequently consulted their physicians. Data quality is also subject to accurate reporting by physicians and patients and may be subject to recall bias. Additionally, patient diagnosis was determined by physician judgment and diagnostic skills rather than a formalized checklist; however, this process is reflective of disease diagnosis in the real world. The high testing rate in Israel may also be due to tests occurring before the illness or during the initial breast cancer diagnosis. In addition, physician-reported mutation testing in blood was used as a proxy for g*BRCA1/2*mut testing. Because blood is used as source material for the testing of circulating tumor DNA, it cannot be verified that all testing conducted on blood samples was germline testing only.

## 5. Conclusions

Substantial percentages of patients with HER2− ABC in the United States and the EU4 do not undergo *BRCA1/2*mut testing, which is important for identifying patients who may benefit from PARPi treatment. Efforts should be made to increase testing rates, especially among older or postmenopausal patients and patients with HR+/HER2− ABC (vs. those with TNBC), without a known *BRCA1/2*-related family history, or who are treated in community medical centers.

## Figures and Tables

**Figure 1 cancers-14-05399-f001:**
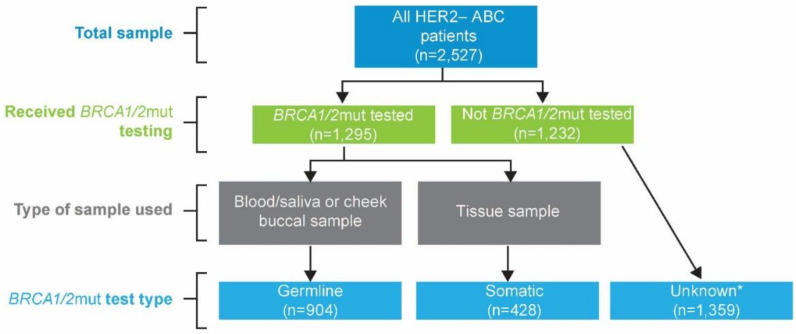
*BRCA1/2* mutation status testing. ABC = advanced breast cancer; *BRCA1/2* = breast cancer susceptibility gene 1 or 2; HER2− = human epidermal growth factor receptor 2 negative. * Includes not tested; not known to have a germline *BRCA1/2* mutation (*BRCA1/2*mut) test result; not known to have *BRCA1/2*mut germline and somatic wildtype test results.

**Figure 2 cancers-14-05399-f002:**
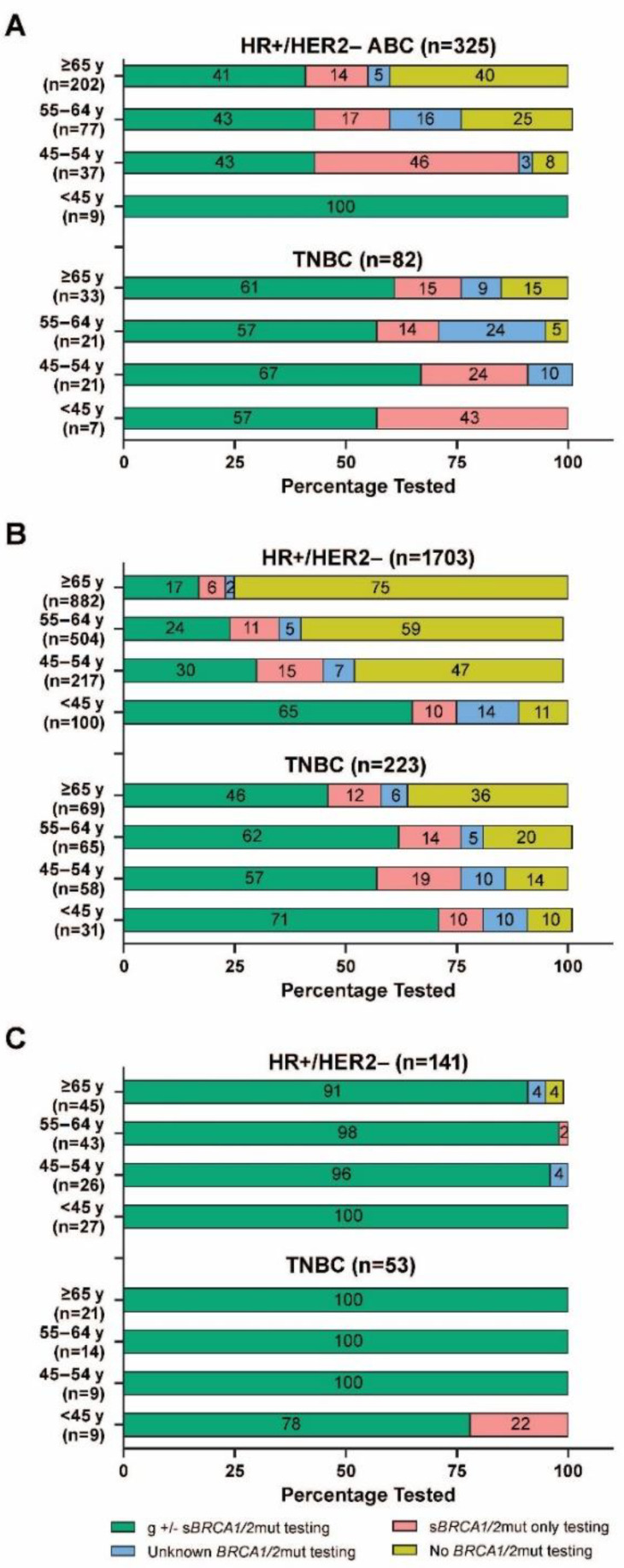
*BRCA1/2*mut testing rates by age group among patients with HER2− ABC in (**A**) the United States, (**B**) the EU4, and (**C**) Israel. Percentages may not add to exactly 100 because of rounding. ABC = advanced breast cancer; *BRCA1/2*mut = breast cancer susceptibility gene 1 or 2 mutation; g = germline; EU4 = European Union 4 (France, Germany, Italy, and Spain); HER2− = human epidermal growth factor receptor 2–negative; HR+ = hormone receptor–positive; s = somatic; TNBC = triple-negative breast cancer.

**Figure 3 cancers-14-05399-f003:**
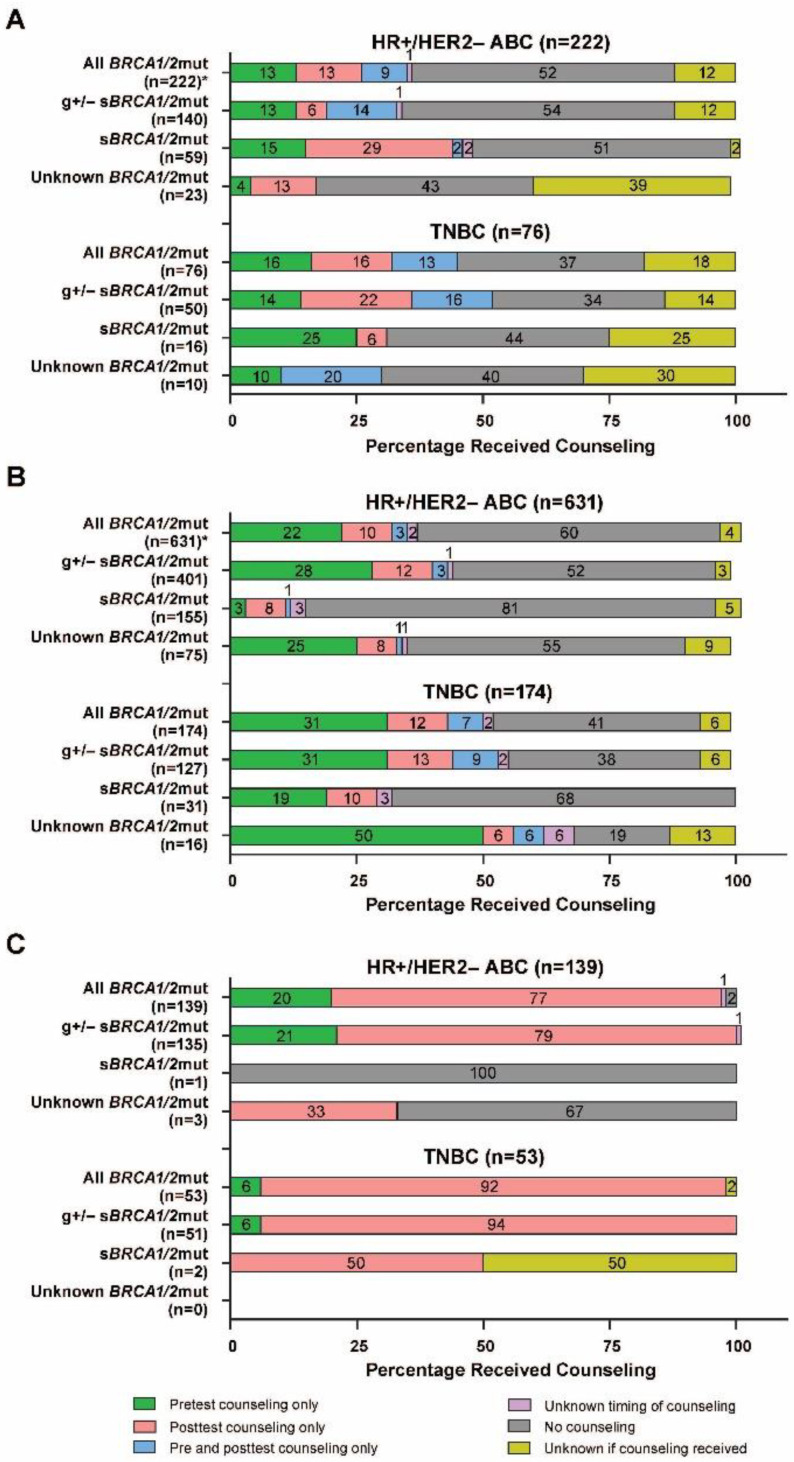
Receipt of genetic counseling by *BRCA1/2*mut testing type among patients with HER2− ABC in (**A**) the United States, (**B**) the EU4, and (**C**) Israel. Percentages may not add to exactly 100 because of rounding. * Indicates a statistically significant difference between those who had and did not have genetic counseling. ABC = advanced breast cancer; *BRCA1/2*mut = breast cancer susceptibility gene 1 or 2 mutation; g = germline; EU4 = European Union 4 (France, Germany, Italy, and Spain); HER2− = human epidermal growth factor receptor 2–negative; HR+ = hormone receptor–positive; s = somatic; TNBC = triple-negative breast cancer.

**Table 1 cancers-14-05399-t001:** Patient demographics and clinical characteristics by *BRCA1/2*mut testing status among patients with HER2—ABC in the United States.

						*p* Value (vs. Not Tested)			
	Any *BRCA1/2*mut Testing (*n* = 298)	g +/− s *BRCA1/2*mut Testing (*n* = 190)	s*BRCA1/2*mut-Only Testing (*n* = 75)	Unknown *BRCA1/2*mut Testing (*n* = 33)	No *BRCA1/2*mut Testing (*n* = 109)	All Tested	g +/− s	s Only	Unknown
Mean patient age, y	62.5	62.9	60.7	64.5	68.9	<0.001	<0.001	<0.001	<0.001
Race									
White/Caucasian	200 (67)	126 (66)	50 (67)	24 (73)	66 (61)	0.240	0.320	0.439	0.223
African American	71 (24)	40 (21)	22 (29)	9 (27)	30 (28)	0.440	0.206	0.868	1.00
Employed	99 (33)	62 (33)	27 (36)	10 (30)	20 (18)	0.003	0.010	0.010	0.151
Premenopausal	24 (8)	13 (7)	10 (14)	1 (3)	2 (2)	0.022	0.095	0.004	0.560
Family history of *BRCA1/2*-related cancer *	56 (19)	36 (19)	15 (20)	5 (15)	7 (6)	0.002	0.003	0.010	0.150
HR status									
HR+/HER2−	222 (74)	140 (74)	59 (79)	23 (70)	103 (94)	<0.001	<0.001	0.002	<0.001
TNBC	76 (26)	50 (26)	16 (21)	10 (30)	6 (6)				
Academic medical center	122 (41)	89 (47)	19 (25)	14 (42)	31 (28)	0.021	0.002	0.737	0.140
Community-based center	176 (59)	101 (53)	56 (75)	19 (58)	78 (72)				

Values are *n* (%) unless noted otherwise. ABC = advanced breast cancer; *BRCA1/2*mut = breast cancer susceptibility gene 1 or 2 mutation; g = germline; HER2− = human epidermal growth factor receptor 2–negative; HR+ = hormone receptor–positive; s = somatic; TNBC = triple-negative breast cancer. * Defined as a family history of breast, ovarian, peritoneal, prostate, pancreatic, gastric, and/or fallopian tube cancer.

**Table 2 cancers-14-05399-t002:** *BRCA1/2*mut testing rates by HR status among patients with HER2– ABC in the United States, the EU4, and Israel.

	United States			EU4			Israel		
	HR+/ HER2− (*n* = 325)	TNBC (*n* = 82)	*p* Value	HR+/ HER2− (*n* = 1703)	TNBC (*n* = 223)	*p* Value	HR+/ HER2− (*n* = 141)	TNBC (*n* = 53)	*p* Value
Any *BRCA1/2*mut testing	222 (68)	76 (93)	<0.001	631 (37)	174 (78)	<0.001	139 (99)	53 (100)	>0.99
g +/− s*BRCA1/2*mut testing	140 (43)	50 (61)	0.004	401 (24)	127 (57)	<0.001	135 (96)	51 (96)	>0.99
s*BRCA1/2*mut-only testing	59 (18)	16 (20)	0.752	155 (9)	31 (14)	0.029	1 (1)	2 (4)	0.182
Unknown *BRCA1/2*mut testing	23 (7)	10 (12)	0.171	75 (4)	16 (7)	0.090	3 (2)	0 (0)	0.563
No *BRCA1/2*mut testing	103 (32)	6 (7)		1072 (63)	49 (22)		2 (1)	0 (0)	

All values are *n* (%). ABC = advanced breast cancer; *BRCA1/2*mut = breast cancer susceptibility gene 1 or 2 mutation; EU4 = European Union 4 (France, Germany, Italy, and Spain); g = germline; HER2− = human epidermal growth factor receptor 2–negative; HR+ = hormone receptor–positive; s = somatic; TNBC = triple-negative breast cancer.

**Table 3 cancers-14-05399-t003:** *BRCA1/2*mut testing rates by practice setting among patients with HER2—ABC in the United States and the EU4.

	United States			EU4		
	Academic	Community	*p* Value	Academic	Community	*p* Value
HR+/HER2−	(*n* = 121)	(*n* = 204)		(*n* = 951)	(*n* = 752)	
Any *BRCA1/2*mut testing	91 (75)	131 (64)	0.048	386 (41)	245 (33)	0.001
g +/− s*BRCA1/2*mut testing	65 (54)	75 (37)	0.004	236 (25)	165 (22)	0.168
s*BRCA1/2*mut-only testing	15 (12)	44 (22)	0.039	109 (11)	46 (6)	<0.001
Unknown *BRCA1/2*mut testing	11 (9)	12 (6)	0.274	41 (4)	34 (5)	0.905
No *BRCA1/2*mut testing	30 (25)	73 (36)		565 (59)	507 (67)	
TNBC	(*n* = 32)	(*n* = 50)		(*n* = 123)	(*n* = 100)	
Any *BRCA1/2*mut testing	31 (97)	45 (90)	0.396	109 (89)	65 (65)	<0.001
g +/− s*BRCA1/2*mut testing	24 (75)	26 (52)	0.063	77 (63)	50 (50)	0.077
s*BRCA1/2*mut-only testing	4 (13)	12 (24)	0.259	25 (20)	6 (6)	0.003
Unknown *BRCA1/2*mut testing	3 (9)	7 (14)	0.733	7 (6)	9 (9)	0.436
No *BRCA1/2*mut testing	1 (3)	5 (10)		14 (11)	35 (35)	

All values are *n* (%). ABC = advanced breast cancer; *BRCA1/2*mut = breast cancer susceptibility gene 1 or 2 mutation; EU4 = European Union 4 (France, Germany, Italy, and Spain); g = germline; HER2− = human epidermal growth factor receptor 2–negative; HR+ = hormone receptor–positive; s = somatic; TNBC = triple-negative breast cancer.

**Table 4 cancers-14-05399-t004:** *BRCA1/2*mut testing rates by family history of *BRCA1/2*-related cancer * among patients with HER2– ABC in the United States, the EU4, and Israel.

	United States			EU4			Israel		
	Family History	No History	*p* Value	Family History	No History	*p* Value	Family History	No History	*p* Value
HR+/HER2−	(*n* = 43)	(*n* = 234)		(*n* = 280)	(*n* = 1356)		(*n* = 101)	(*n* = 39)	
Any *BRCA1/2*mut testing	36 (84)	156 (67)	0.030	173 (62)	437 (32)	<0.001	100 (99)	38 (97)	0.481
g +/− s*BRCA1/2*mut testing	22 (51)	99 (42)	0.317	120 (43)	274 (20)	<0.001	98 (97)	37 (95)	0.618
s*BRCA1/2*mut-only testing	11 (26)	45 (19)	0.408	33 (12)	111 (8)	0.063	1 (1)	0 (0)	1.00
Unknown *BRCA1/2*mut testing	3 (7)	12 (5)	0.711	20 (7)	52 (4)	0.024	1 (1)	1 (3)	0.481
No *BRCA1/2*mut testing	7 (16)	78 (33)		107 (38)	919 (68)		1 (1)	1 (3)	
TNBC	(*n* = 20)	(*n* = 57)		(*n* = 57)	(*n* = 157)		(*n* = 30)	(*n* = 20)	
Any *BRCA1/2*mut testing	20 (100)	53 (93)	0.568	51 (89)	118 (75)	0.023	30 (100)	20 (100)	1.00
g +/− s*BRCA1/2*mut testing	14 (70)	35 (61)	0.594	38 (67)	86 (55)	0.158	30 (100)	20 (100)	1.00
s*BRCA1/2*mut-only testing	4 (20)	12 (21)	1.00	6 (11)	24 (15)	0.505	0 (0)	0 (0)	1.00
Unknown *BRCA1/2*mut testing	2 (10)	6 (11)	1.00	7 (12)	8 (5)	0.125	0 (0)	0 (0)	1.00
No *BRCA1/2*mut testing	0 (0)	4 (7)		6 (11)	39 (25)		0 (0)	0 (0)	

All values are *n* (%). ABC = advanced breast cancer; *BRCA1/2*mut = breast cancer susceptibility gene 1 or 2 mutation; EU4 = European Union 4 (France, Germany, Italy, and Spain); g = germline; HER2− = human epidermal growth factor receptor 2–negative; HR+ = hormone receptor –positive; s = somatic; TNBC = triple-negative breast cancer. * Defined as a family history of breast, ovarian, peritoneal, prostate, pancreatic, gastric, and/or fallopian tube cancer.

**Table 5 cancers-14-05399-t005:** Patient demographics and clinical characteristics by *BRCA1/2*mut testing status among patients with HER2– ABC in the EU4.

						*p* Value (vs. Not Tested)			
	Any *BRCA1/2*mut Testing (*n* = 805)	g +/− s*BRCA1/2*mut Testing (*n* = 528)	s*BRCA1/2*mut-Only Testing (*n* = 186)	Unknown *BRCA1/2*mut Testing (*n* = 91)	No *BRCA1/2*mut Testing (*n* = 1121)	All Tested	g +/− s	s Only	Unknown
Mean patient age, y	58.0	57.9	59.7	55.3	66.7	<0.001	<0.001	<0.001	<0.001
Race									
White/Caucasian	752 (93)	489 (93)	180 (97)	83 (91)	1063 (95)	0.199	0.092	0.357	0.148
Employed	206 (26)	124 (23)	51 (27)	31 (34)	124 (11)	<0.001	<0.001	<0.001	<0.001
Premenopausal	118 (15)	75 (14)	25 (14)	18 (20)	33 (3)	<0.001	<0.001	<0.001	<0.001
Family history of *BRCA1/2*-related cancer *	224 (28)	158 (30)	39 (21)	27 (30)	113 (10)	<0.001	<0.001	<0.001	<0.001
HR status									
HR+/HER2−	631 (78)	401 (76)	155 (83)	75 (82)	1072 (96)	<0.001	<0.001	<0.001	<0.001
TNBC	174 (22)	127 (24)	31 (17)	16 (18)	49 (4)				
Academic medical center	495 (61)	313 (59)	134 (72)	48 (53)	579 (52)	<0.001	0.004	<0.001	0.913
Community-based center	310 (39)	215 (41)	52 (28)	43 (47)	542 (48)				

Values are *n* (%) unless noted otherwise. ABC = advanced breast cancer; *BRCA1/2*mut = breast cancer susceptibility gene 1 or 2 mutation; EU4 = European Union 4 (France, Germany, Italy, and Spain); g = germline; HER2− = human epidermal growth factor receptor 2–negative; HR+ = hormone receptor–positive; s = somatic; TNBC = triple-negative breast cancer. * Defined as a family history of breast, ovarian, peritoneal, prostate, pancreatic, gastric, and/or fallopian tube cancer.

## Data Availability

Data collection was undertaken by Adelphi Real World as part of an Adelphi DSP independent survey sponsored by multiple pharmaceutical companies, of which one was Pfizer Inc. Publication of study results was not contingent on the sponsor’s approval or censorship of the manuscript. All data that support the findings of this study are the intellectual property of Adelphi Real World. All requests for access should be addressed directly to Katie Lewis at katie.lewis@adelphigroup.com.

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
