# Peer review of "BRCA1/2 Mutation Testing in Patients with HER2-Negative Advanced Breast Cancer: Real-World Data from the United States, Europe, and Israel"

_cancers, 2022, doi:10.3390/cancers14215399_

Round 1
Reviewer 1 Report
The paper from Mahtani et al. reports the different frequency of testing for BRCA1 and BRCA2 germline mutations in patients with advanced breast cancer and the clinical characteristics of tested patients, comparing Europe, Israel and the United States.
The study shows that in Europe and the United States the percentage of tested patients with advanced breast cancer is still low compared to recent guidelines, which recommend testing all patients with advanced breast cancer for taking advantage of PARP inhibitor treatment in case of germline mutation presence.
In my opinion, although it makes sense to present this data, the interest of this paper for readers remains limited, since every clinician in Europe or the USA knows that not all patients with advanced breast cancer are tested for germline BRCA1/2 mutation yet, and it might be more interesting to investigate the reasons why this happens.
The study is written in proper English and the presentation of results is quite clear.
I suggest some minor revisions in this paper prior to accepting it for publication.
Minor comments:
- one might comment on why among patients in the United States and Europe, so few are premenopausal (8% vs. 27% of those in Israel): certainly, this affects the percentage of patients tested: I suggest to mention this in the discussion paragraph;
- paragraph statistical analysis: add the non-abbreviated form for “SD”;
- paragraph discussion: please correct ESMO “European Society for Medical Oncology” instead of “Molecular Oncology”; please rephrase this sentence “guidelines on testing for gBRCA1/2mut have expanded to include new therapeutic indications in addition to a strong family history”: saying that only a strong family history is a criterion to test patients is too imprecise and too general.
Author Response
The paper from Mahtani et al. reports the different frequency of testing for BRCA1 and BRCA2 germline mutations in patients with advanced breast cancer and the clinical characteristics of tested patients, comparing Europe, Israel and the United States. The study shows that in Europe and the United States the percentage of tested patients with advanced breast cancer is still low compared to recent guidelines, which recommend testing all patients with advanced breast cancer for taking advantage of PARP inhibitor treatment in case of germline mutation presence. In my opinion, although it makes sense to present this data, the interest of this paper for readers remains limited, since every clinician in Europe or the USA knows that not all patients with advanced breast cancer are tested for germline BRCA1/2 mutation yet, and it might be more interesting to investigate the reasons why this happens. The study is written in proper English and the presentation of results is quite clear. I suggest some minor revisions in this paper prior to accepting it for publication.
Minor comments:
-one might comment on why among patients in the United States and Europe, so few are premenopausal (8% vs. 27% of those in Israel): certainly, this affects the percentage of patients tested: I suggest to mention this in the discussion paragraph;
Response: The following has been added to the Discussion: “In addition, the proportion of patients that were premenopausal was much higher in Israel (27%) compared to the United States (6%) and EU4 (8%). An awareness that breast cancer incidence is higher in younger patients (ie, premenopausal patients) that carry a BRCA1/2mut may have also contributed to the high rate of BRCA1/2mut testing in Israel.”
- paragraph statistical analysis: add the non-abbreviated form for “SD”;
Response: SD has been spelled out as suggested.
- paragraph discussion: please correct ESMO “European Society for Medical Oncology” instead of “Molecular Oncology”;
Response: Correction has been made.
Please rephrase this sentence “guidelines on testing for gBRCA1/2mut have expanded to include new therapeutic indications in addition to a strong family history”: saying that only a strong family history is a criterion to test patients is too imprecise and too general.
Response: The sentence has been revised to, “…, guidelines on testing for gBRCA1/2mut have expanded to include new therapeutic indications in addition to clinical criteria such as patients diagnosed at an early age and patients with a strong family history (eg, a first-degree relative diagnosed with breast cancer at an early age or with TNBC, two or more close relatives with breast cancer at any age, two or more close blood relatives with breast, pancreatic or prostate cancer at any age or a known BRCA1/2 mutation in the family)”.
Reviewer 2 Report
In this study the authors analysed differences in BRCA1/2 testing within different countries. The authors should present their results in a more concise manner. Some reported data do not add quality to the study and in some cases are unnecessary.
The most significant data (family history and HR status in USA and EU4) need to be better underlined, eliminating some not significant results.
For example, in table 1 there is the line “Female”. Almost all patients, as expected for breast cancer, are females, so the data repeated almost exactly the overall data.
The table with the data from Israel has not significant results because almost all the patients were BRCA tested. Thus, what is the meaning of this table? In the same table the number of “employed” is very low, I guess because no information was recovered.
Author Response
In this study the authors analysed differences in BRCA1/2 testing within different countries. The authors should present their results in a more concise manner. Some reported data do not add quality to the study and in some cases are unnecessary. The most significant data (family history and HR status in USA and EU4) need to be better underlined, eliminating some not significant results.
For example, in table 1 there is the line “Female”. Almost all patients, as expected for breast cancer, are females, so the data repeated almost exactly the overall data. The table with the data from Israel has not significant results because almost all the patients were BRCA tested. Thus, what is the meaning of this table? In the same table the number of “employed” is very low, I guess because no information was recovered.
Response: As suggested by the reviewer, the following nonsignificant results have been removed from the manuscript: Table 6 (Patient Demographics and Clinical Characteristics by BRCA1/2mut Testing Status Among Patients With HER2− ABC in Israel) and Female gender data from Tables 1 and 5.
Reviewer 3 Report
The manuscript by Mahtani et al. analyzed the differences in demographics, clinical characteristics, and BRCA1/2mu testing in breast cancer patients within the United States, Europe, and Israel. The manuscript is well-written and provides useful information on BRCA1/2mut testing rates and related characteristics among patients in different counties. I only have a few minor questions as described below.
1. The manuscript performed a thorough analysis of BRCA1/2mu testing rates. I am wondering whether a higher BRCA1/2mu testing rate will result in better overall survival.
2. Page 8 “73 [91%] from a genetic counselor and 8 [10%] from the treating physician”. I cannot find this in figure 3.
Author Response
The manuscript by Mahtani et al. analyzed the differences in demographics, clinical characteristics, and BRCA1/2mu testing in breast cancer patients within the United States, Europe, and Israel. The manuscript is well-written and provides useful information on BRCA1/2mut testing rates and related characteristics among patients in different counties. I only have a few minor questions as described below.
- The manuscript performed a thorough analysis of BRCA1/2mu testing rates. I am wondering whether a higher BRCA1/2mu testing rate will result in better overall survival.
Response: This is an interesting question; however, we are unable to evaluate this as survival data was not captured in the current study. The authors are unaware of any studies that have evaluated associations of BRCA1/2mut testing rates and survival.
- Page 8 “73 [91%] from a genetic counselor and 8 [10%] from the treating physician”. I cannot find this in figure 3.
Response: The breakdown of who the 36% of HR+/HER2- patients that received counseling from (ie, 73 [91%] from a genetic counselor and 8 [10%] from the treating physician) is not shown in Figure 3, this data is only presented in the text of Results. Figure 3 only presents the proportion of patients who received counseling and the type of counseling received.